# Microstructures and Accuracy of Graph Recall by Large Language Models

**Yanbang Wang**
Cornell University
ywangdr@cs.cornell.edu

**Hejie Cui**
Stanford University
hejie.cui@stanford.edu

**Jon Kleinberg**
Cornell University
kleinberg@cornell.edu

## Abstract

Graphs data is crucial for many applications, and much of it exists in the relations described in textual format. As a result, being able to accurately recall and encode a graph described in earlier text is a basic yet pivotal ability that large language models (LLMs) need to demonstrate if they are to perform reasoning tasks that involve graph-structured information. Human performance at graph recall by has been studied by cognitive scientists for decades, and has been found to often exhibit certain structural patterns of bias that align with human handling of social relationships. To date, however, we know little about how LLMs behave in analogous graph recall tasks: do their recalled graphs also exhibit certain biased patterns, and if so, how do they compare with humans and affect other graph reasoning tasks? In this work, we perform the first systematical study of graph recall by LLMs, investigating the accuracy and biased microstructures (local subgraph patterns) in their recall. We find that LLMs not only underperform often in graph recall, but also tend to favor more triangles and alternating 2-paths. Moreover, we find that more advanced LLMs have a striking dependence on the domain that a real-world graph comes from — by yielding the best recall accuracy when the graph is narrated in a language style consistent with its original domain.

## 1 Introduction

Large language models (LLMs) achieve remarkable progress in recent years. In many applications, LLMs' success relies on appropriate handing of graph-structured information embedded in text. For example, to accurately answer questions pertaining to the characters in a story, it is crucial for an LLM to be able to recognize and analyze the social network of relations among these characters. In fact, graph-structured information is ubiquitous across many language-based applications, such as structured commonsense reasoning [32], multi-agent communications [3], multi-hop question answering [15], and more. LLMs' graph reasoning ability has thus become an active research topic.

Existing works on LLMs' graph reasoning ability have been primarily posited in the context of various graph tasks, from most basic ones such as computing node degree, graph diameter, clustering coefficient, or checking cycles [48, 20, 18], to more challenging ones such as node/graph classification [14, 38] and link-based recommendations [55]. Sec. 7 provides a more comprehensive survey.

But all of the tasks above rely on the premise that an LLM is able to start from the graph that is described in the text it is given. Thus, our key starting observation here is that all of these tasks rely on a pivotal (and perhaps seemingly trivial) ability of LLMs – to recall and encode a set of relations described in earlier text. In this paper, we consider a graph recall task which has been extensively studied in cognitive science [8, 44, 7, 9, 41], and which formalizes this basic goal, illustrated in Figure 1: a set of pairwise relationships is described in a simple narrative form to the experimental subject (human or LLM); then at a later point in the experiment, the subject is asked to recall and describe these relationships explicitly in the form of a graph.

38th Conference on Neural Information Processing Systems (NeurIPS 2024).

The rationale for studying an LLM's graph recall is simple. If an LLM cannot even accurately recall the graph it is asked to reason upon, it will not be able to do well in any of the more advanced graph tasks surveyed above. Further, structural patterns in recall errors may propagate to (or even serve as the basis for) the behavior of LLMs in these more complex graph tasks. Therefore, we consider it important to investigate LLMs' graph recall ability, a topic absent from existing literature. The edge prediction task is related to but fundamentally different from the graph recall task: the correct answer for graph recall always exists in the prompt and can be directly extracted, which is not true for any prediction tasks.

Meanwhile, the existing two decades of studies on human's graph recall ability provide another fascinating perspective to motivate our study. Cognitive scientists have found through substantial human experiments that, when memorizing a social network, humans extensively employ certain compression heuristics, such as triadic closure, near-clique completion, and certain degree biases [8, 35], due to natural limits on cognitive capacity. Further studies show that a person's ability to accurately recall a social network, along with the microstructures (local subgraph patterns, or network motifs [2, 34]) in their recalled network, not only has profound influence on their social decisions [25, 11, 52], but also varies on different styles of graph narrative [44], and the sex of the person [7, 22]. Do LLMs use similar compression heuristics as humans, and how are they affected by different working contexts? As LLMs become increasingly integrated into various social applications, it becomes crucial to understand LLM's behaviors and their associations with human behaviors in these regards.

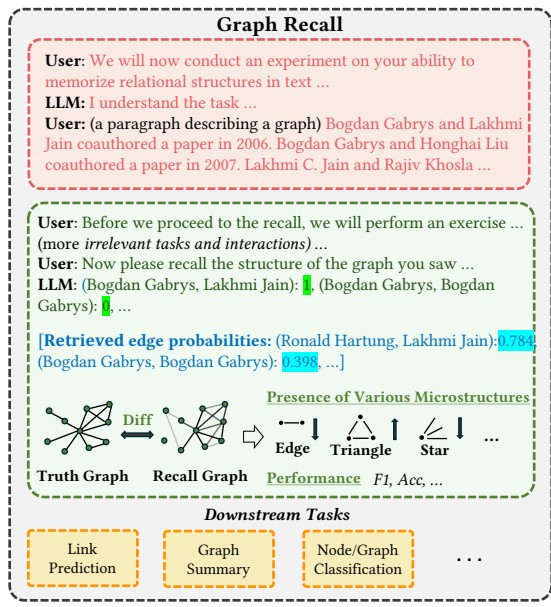

Figure 1: Graph recall is a simple task but also a crucial pivot for other graph reasoning tasks.

The human cognition studies not only motivate our study, but also establish a scientific foundation for our experimental designs. Similar to [42] which uses political orientation tests designed for humans to assess LLM's political bias, we also find some of the protocols employed by [8] for testing human's graph recall highly instructional, including: (1) memory clearance, where a classical word span test [17] is conducted between the presentation of the graph content and the query prompt; this serves as a chat buffer that helps simulate the delayed queries in many real-world applications; (2) analyzing biased patterns in graph recall via Exponential Random Graph Model (ERGM), a probabilistic generalization to the network motif methods in graph mining; (3) focusing on the probability of the tokens in subject's response, rather than just their presence; we replicate this through the `log_prob` parameter in GPT series or conducting Monto Carlo sampling for Gemini. We will elaborate on these in the following sections.

**Our Work** In this work, we investigate the several primary questions regarding the capabilities of LLMs in recalling graph structures. First, we examine the accuracy and microstructures in graphs recalled by LLMs through experiments on real-world graph structures, as well as compare these results with human performance. Second, we explore factors affecting LLMs' graph recall abilities, focusing on memory clearance strength, and narrative styles of graph encoding, which are known to affect human's graph recall. Finally, we also consider how LLMs' graph recall influences their performance in downstream tasks like link prediction, and discuss actionable insights that our findings provide for future research. To summarize, our work makes the following contributions:

1. We propose graph recall as a simple yet fundamental task for understanding LLM's graph reasoning abilities, drawing its connection with the existing cognitive studies on human's graph recall ability.

2. We are the first to design and conduct systematical studies on the accuracy and biased microstructures of LLM's graph recall, and to compare the results with humans.

3. We present many important and interesting findings on LLM's behaviors in graph recall, which significantly helps deepen our understandings about LLM's graph reasoning ability.

## 2  Preliminaries

### 2.1  Exponential Random Graph Model (ERGM)

We use the Exponential Random Graph Model (ERGM) [39] to characterize the statistical significance of the various microstructural patterns ("network motifs" [2, 34]) in the recalled graphs. ERGM is a special case of the exponential family dedicated for modeling graph-structured data.

Formally, let $\mathcal{G}$ be the probability space of all possible graphs over $n$ nodes, and $G = (V, E, f) \in \mathcal{G}$ be a random (graph) variable, where $V$ is the set of nodes, $E$ is the set of edges in $G$, and $f : E \to [0, 1]$ is a edge probability function. Assuming independence between edges, the probability of $G$ can be written as:

$$P(G) = \prod_{e \in E} f(e) \prod_{e \notin E} (1 - f(e)) \tag{1}$$

$A$ is a list of $k$ predefined microstructural patterns in $G$. Fig.2's Step 6 shows $k = 5$ such patterns that we primarily investigate in this work, following [8]. The conditional probability of observing $G$ given the parameter vector $\theta$ of length $k$ is defined to be

$$P(G|\theta) = \frac{\exp\{\sum_{i=1}^{k} \theta_i s_i(G)\}}{c_\theta} = \frac{\exp\{\theta^T \mathbf{s}(G)\}}{c_\theta} \tag{2}$$

where $\mathbf{s}(G)$ is the sufficient statistics of $G$, and each $\mathbf{s}_i(G)$ is a count of the number of occurrences of $A[i]$ in $G$; $c_\theta$ is the normalizing constant that only depends on $\theta$. We assume an uninformative prior for $\theta \sim [-10, 10]$. The posterior $P(\theta|G)$ can then be optimized via MAP under Bayesian framework. $\theta$ measures the strength of presence of the $k$ microstructural patterns we care about. A large $\theta_i$ means a strong presence of pattern $A[i]$ in $G$.

### 2.2  Memory Clearance

The word span test [17] is a standard method for measuring a human's working memory capacity. The test requires the subject to read a series of sentence sets out loud, and then recall the last word in each previous sentence in the current set. The number of sentences in each set gradually increases from two to seven, *i.e.* from three sets of two sentences to three sets of seven sentences. The test continues until the subject fails to recall the final words for two out of three sets of a given size. See Appendix B.2.1 for the sentence sets we used.

Many cognition studies [8, 7, 6, 5] have adopted this test in their experiment to (1) serve as a chat buffer or spoiler that simulates the delayed query in real-world applications, and (2) clear the short-term memory of the subject, which allows researchers to better focus on relatively persistent patterns in memory structures. It is important to note that transformer-based LLMs are stateless models that, in strict sense, do ***not*** have memory. Nevertheless, we choose to preserve the (slightly misleading) term "memory clearance" to stay aligned with literature and to emphasize the close analogy and behavioral resemblance between LLMs and humans in graph recall.

## 3  Microstructures and Accuracy of Graph Recall by LLMs

Being able to correctly recall a graph described in earlier text is a fundamental ability for LLMs to perform graph reasoning. While graph recall may seem an easy task for the high-capable LLMs nowadays, our extensive experiment shows that their performance is in fact far from perfect.

This section presents our study on the performance and the microstructures of graph recall by LLMs. Our experimental protocols are introduced in Sec.3.1, followed by Sec.3.3 which presents head-to-head comparisons of LLMs and humans under [8]'s framework. Sec.3.2 substantiates the analysis by experiments on more diverse datasets. Our **code and data** are reported in Appendix A.

**LLMs Tested:** GPT-3.5 [10], GPT-4 [1], Gemini-Pro [46]. We also examined Llama 2 (13B) [47], but they can rarely follow through on our instructions.

### 3.1  Experimental Protocols and Datasets

Our staged protocols are visualized in Fig.2 and explained below. To recall each graph sample, an LLM needs to separately go through the entire pipeline. The stages proceed in an auto-regressive manner. In other words, the LLM's intermediate response is always appended to the current thread as additional conservation context, before we proceed to the next stage along the pipeline.

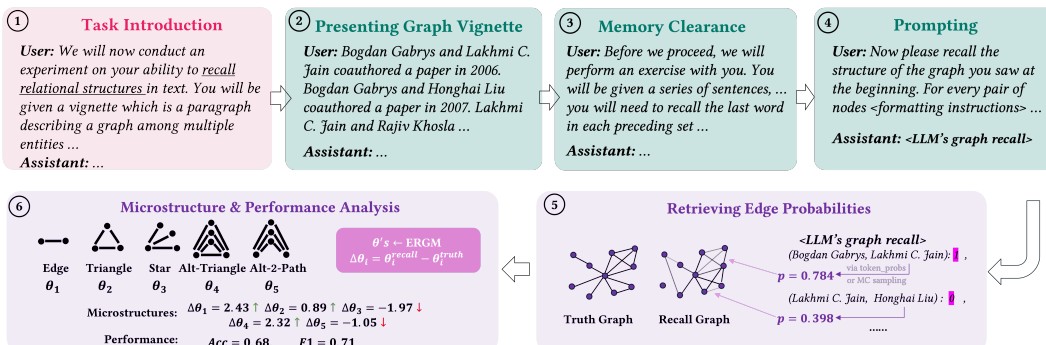

Figure 2: Experimental protocols for analyzing microstructures and accuracy of LLM's graph recall. See Sec.3.1 for detailed explanations.

**Step 1: Task Introduction.** The LLM is informed that this is a graph recall test, and that the recall task may happen at a later time. It is also incentivized to yield its best performance. These components follow the ones used in [8] for human studies.

**Step 2: Presenting Graph Vignette.** A vignette in sociology is a short, descriptive story that encodes the central piece of information for soliciting subject's response. Here, our vignette encodes the graph structures sampled from a certain application domain using a certain narrative style — see the dataset tab below for details.

**Step 3: Memory Clearance.** A standard word span test [17] is conducted with the LLM. See Sec.2.2 for details. We use the same set of sentences as in [8].

**Step 4: Prompting.** We use zero-shot prompting with moderate formatting instructions for answers. This follows both [8]'s protocol and the finding in [18] that simple prompts are the best for simple tasks, which we also empirically observed.

**Step 5: Retrieving Edge Probabilities.** We are interested in both the existence and the probability of each edge (*i.e.* the $f$ in ERGM) in graphs recalled by LLM – the latter lets us examine LLM's behavior at finer granularities. We use two tricks to retrieve edge probabilities from LLM's answers:

- For GPT series, we can directly access token probabilities through the `ChatCompletion` API. We instruct the model to output $1$ for each edge it believes to exist, and $0$ otherwise. Then, we retrieve and normalize the probabilities for tokens $0$ and $1$, using the latter as the edge probability.

- For Gemini-Pro whose token probabilities are not accessible via the API, we conduct Monte Carlo sampling for each potential edge (repeating the query for $100$ times), and use the fraction of existence as the edge probability.

Note that the retrieved edge probabilities essentially constitute a probability graph.

**Step 6: Microstructure Analysis & Performance Measurement.** We use the ERGM introduced in Sec. 2.1 to model both the recalled graph and the ground truth, in order to reveal statistically significant structural patterns, or "microstructures" as termed by [8], in the recalled graphs. Specifically, for each microstructural pattern, we compute the gap between its estimated coefficient on the recalled graph against its estimated coefficient on the ground-truth graph. Steps 1 - 6 are repeated for 100 different graphs uniformly sampled from the same domain to compute confidence intervals.

**Datasets.** We create five graph datasets from the following application domains. (1) Co-authorship: DBLP (1995-2005); (2) Social network: Facebook [27]; (3) Geological network: CA road; (4) Protein interactions: Reactome [16]; (5) Erdős–Rényi graph: as in [18]. Each dataset comprises of 100 graphs and their corresponding textual descriptions in the domain's narrative language, which are generated from templates that are intentionally kept simple: Fig.2's Step 2 shows an example for the DBLP dataset. More examples and details are provided in Appendix B.

Graphs in the first four datasets are uniformly sampled as ego-network, with the central node removed so that no graph isomorphism gets excluded by the sampling scheme. Each graph has 5 to 30 nodes. The Erdős–Rényi graph are generated by uniformly sampling a $p$ value from $[0, 1]$. Also note that because the templates used are simple and fixed, we can easily create baselines by reverse-engineering the template to achieve *perfect* accuracy in graph recall.

Table 1: Microstructural patterns and performance of graph recall by LLMs on graphs sampled from various application domains; mean $\pm$ ci$_{95\%}$ reported. The numbers reported for microstructural patterns are signfance parameters $\theta$ computed by the ERGM model introduced in Sec. 2.1. A positive (negative) number means the LLM is biased towards encouraging (depressing) the corresponding microstructural pattern in recalled graphs. The patterns are visualized in Fig.2 Step 6. Full table available in Appendix C.

| Model / Dataset | Microstructure | | | | | Performance | |
| --- | --- | --- | --- | --- | --- | --- | --- |
| | *Edge* | *Triangle* | *Star* | *Alt-Triangle* | *Alt-2-Path* | Accuracy (%) | F1 (%) |
| **GPT-3.5** | | | | | | | |
| Facebook | -3.70 ± 5.80 | 1.72 ± 1.34 ↑ | -0.70 ± 3.00 | -0.91 ± 2.25 | 3.31 ± 1.77 ↑ | 71.60 ± 4.07 | 72.34 ± 3.54 |
| CA Road | 0.64 ± 0.91 | 7.31 ± 3.49 ↑ | -3.46 ± 1.77 ↓ | -1.47 ± 0.92 ↓ | 2.35 ± 1.29 ↑ | 95.52 ± 2.67 | 92.95 ± 2.95 |
| Reactome | -18.01 ± 6.22 ↓ | -0.71 ± 4.62 | 4.96 ± 3.81 ↑ | -6.32 ± 4.23 ↓ | 4.43 ± 4.93 | 53.68 ± 3.32 | 44.47 ± 6.19 |
| DBLP | -8.12 ± 3.25 ↓ | 1.17 ± 4.43 | 7.17 ± 2.44 ↑ | -11.16 ± 4.35 ↓ | 5.77 ± 3.76 ↑ | 72.47 ± 3.61 | 65.08 ± 4.73 |
| Erdős–Rényi | -1.40 ± 5.01 | 9.57 ± 4.89 ↑ | 1.40 ± 2.71 | -0.41 ± 4.19 | 3.36 ± 2.76 ↑ | 55.20 ± 3.32 | 49.49 ± 5.29 |
| **GPT-4** | | | | | | | |
| Facebook | -0.17 ± 0.50 | 0.05 ± 0.78 | 0.06 ± 0.21 | -0.01 ± 0.26 | 0.01 ± 0.06 | 99.80 ± 0.11 | 99.75 ± 0.13 |
| CA Road | 1.34 ± 1.62 | 6.07 ± 3.93 ↑ | -3.09 ± 2.08 ↓ | -1.27 ± 0.96 ↓ | 1.85 ± 0.91 ↑ | 98.11 ± 2.54 | 98.00 ± 2.74 |
| Reactome | -11.54 ± 5.82 ↓ | 0.82 ± 4.28 | 2.69 ± 2.87 | -1.99 ± 2.03 | -0.46 ± 2.14 | 77.04 ± 4.13 | 76.80 ± 4.59 |
| DBLP | -1.26 ± 2.56 | -1.41 ± 3.26 | -0.69 ± 3.74 | 1.71 ± 2.96 | -1.59 ± 2.24 | 98.70 ± 0.74 | 97.88 ± 1.65 |
| Erdős–Rényi | -1.49 ± 4.15 | 1.95 ± 1.54 ↑ | 0.52 ± 2.79 | -1.26 ± 1.66 | -0.45 ± 0.76 | 60.34 ± 2.37 | 44.03 ± 5.20 |
| **Gemini-Pro** | | | | | | | |
| Facebook | -2.31 ± 1.26 ↓ | -2.29 ± 2.99 | 1.50 ± 2.37 | 2.40 ± 1.37 ↑ | 0.67 ± 1.10 | 51.13 ± 2.58 | 34.56 ± 4.72 |
| CA Road | 0.84 ± 1.68 | 2.02 ± 0.29 ↑ | 5.24 ± 0.27 ↑ | 0.89 ± 0.28 ↑ | 6.82 ± 0.62 ↑ | 44.69 ± 3.86 | 31.92 ± 4.83 |
| Reactome | -11.94 ± 4.65 ↓ | 1.27 ± 5.22 | 13.47 ± 4.70 ↑ | 3.32 ± 4.57 | 4.15 ± 4.01 ↑ | 54.09 ± 2.92 | 46.59 ± 5.90 |
| DBLP | -19.47 ± 3.30 ↓ | -1.72 ± 3.45 | 11.24 ± 2.45 ↑ | -11.08 ± 3.90 ↓ | 10.83 ± 4.53 ↑ | 46.33 ± 3.54 | 47.36 ± 4.55 |
| Erdős–Rényi | -2.86 ± 5.16 | 1.51 ± 5.04 | -0.66 ± 3.95 | 0.59 ± 2.81 | -0.70 ± 1.15 | 52.40 ± 4.41 | 22.27 ± 4.64 |

## 3.2 Results and Analysis

Table 1 shows the results of microstructural patterns and performance of LLMs in our graph recall test. We primarily investigate the five microstructural patterns as shown in Fig. 2. This is because these patterns have been observed to be biased patterns in human studies and are substantiated with rich sociological explanations [8]; other patterns may also be interesting to examine though, which we leave for future work. A positive/negative value means the LLM is biased towards encouraging/depressing the corresponding microstructural pattern in recalled graphs. We have the following findings.

**LLMs underperform in the graph recall test.** We start by examining the performance metrics on the right columns. None of the models are able to perform perfectly on any dataset — in fact not even close in most cases. The unsaturated performance shows that the graph recall test is a meaningful task to investigate. The result also helps partially explain the poor performance of LLMs on many other graph tasks including node degree, edge count, and cycle check [48, 18, 20].

**LLMs may tend to forget, rather than hallucinate edges.** The "edge" column shows an interesting result that LLMs generally recall fewer edges than the ground truth. This crucially tells us that LLM's bias in other microstructural patterns may more likely be the consequence of *selective forgetting*, rather than *hallucination*.

**An LLM's microstructural bias is relatively robust.** For each model, the colors in each column are relatively consistent. This means that an LLM may have have some relatively persistent bias in its microstructual patterns, which does not change significantly across different application domains. This is a positive indicator of the generalizability of our findings above.

## 3.3 LLMs Compared with Humans in Graph Recall

Brashears et al. reported experimental results of social network recall on a total of 301 humans [8]. Here we report how we build upon this existing result to compare LLMs' and humans' behaviors in the social network recall test.

To stay aligned with Brashears' settings, we use their dataset which consists of two 15-node social networks: an *irreducible* one which contains no cliques, and a *reducible* one which contains multiple cliques. Appendix B introduces more details. Table 2 shows the comparison results.

**Regarding the types of bias (forgetting vs. hallucination), LLMs are relatively consistent with humans.** Both LLMs and humans tend to forget *edges* and *alt-triangles* while "hallucinating"

*triangles* and *stars*. This interesting finding may reveal that LLMs have some information decoding and recall mechanisms similar to human memory.

**Regarding the strength of the bias, LLMs tend to have weaker forgetting but stronger hallucinations than humans.** On microstructures such as *edges* and *alt-triangles*, LLMs tend to have a weaker forgetting pattern, while on some prominently increasing microstructures (e.g., *triangles*), LLMs tend to demonstrate strong hallucinating patterns. These findings suggest that while LLMs may have a distinct advantage than humans in retaining certain graph structures in their memory, they may still struggle with accurately recalling, potentially limiting their ability to perform complex or critical graph tasks.

Table 2: LLMs vs. Humans: microstructural patterns and performance of graph recall, conducted on the two social networks used in [8]. Numbers on the "humans" row were taken from Brashears' paper and postprocessed by us. * not reported in Brashears' paper.

| Model | Microstructure | | | | | Performance |
| | Edge | Triangle | Star | Alt-Triangle | Alt-2-Path | Accuracy (%) |
|---|---|---|---|---|---|---|
| "Irreducible" social network | | | | | | |
| Humans | -3.19+-5.97 | 9.52+-1.23 ↑ | 2.31+-0.11 ↑ | -3.39+-1.22 ↓ | -1.71+-3.32 | 29.72 |
| GPT-3.5 | -2.23+-2.79 | 5.40+-0.51 ↑ | 4.69+-2.09 ↑ | -1.74+-0.87 ↓ | -1.26+-1.39 | 31.51 |
| GPT-4 | -5.36+-1.59 ↓ | 11.40+-3.07 ↑ | 3.40+-1.45 ↑ | -2.05+-1.22 ↓ | -0.96+-0.48 ↓ | **95.71** |
| Gemini-Pro | 9.82+-0.28 ↑ | 7.63+-0.45 ↑ | -0.46+-1.12 ↑ | -2.88+-0.98 ↓ | -0.47+-0.99 | 24.99 |
| "Reducible" social network | | | | | | |
| Humans | -9.41+-2.21 ↓ | 1.71+-0.51 ↑ | -1.67+-0.03 ↓ | —* | 4.34+-1.72 ↑ | 17.80 |
| GPT-3.5 | -5.64+-1.56 ↓ | -0.13+-5.71 | -2.52+-16.43 | -4.79+-14.96 | 1.91+-2.97 | 51.11 |
| GPT-4 | -2.82+-2.51 ↓ | -1.10+-0.86 ↓ | -1.70+-0.97 ↓ | -0.93+-0.64 ↓ | 0.47+-0.27 | **95.74** |
| Gemini-Pro | -3.25+-2.09 ↓ | 10.92+3.52 ↑ | -1.99+-8.35 | -0.44+-4.37 | 0.03+-0.39 | 38.82 |

# 4 What Affects LLM's Graph Recall?

It is a natural question to ask about factors that can affect an LLM's performance in the graph recall task. Many existing works have investigated graph properties and prompting methods as two key variants affecting LLM's performance in graph reasoning tasks [48, 14]. Here we focus on several interesting factors that are less explored but still play a crucial role in graph recall: narrative style, strength of memory clearance, and sex priming (Appendix D).

## 4.1 Narrative Style

**Motivation.** The effect of narrative styles on several graph reasoning abilities has been studied by [18] with synthetic random graphs. Here we will present an interesting finding on real-world graphs, and by novelly cross-evaluating narrative styles and real-world domains.

The key idea is the following: it is known that graphs sampled from different domains have different *distributions of topology*, e.g. road networks are usually star-shaped, whereas social networks usually have more triangles [45]. Meanwhile, each domain has its own *style of narration*: for describing road networks geographical locations and names are often used, while for describing social networks names and personal relationships are used more often. Our experiment thus far has always used for each dataset the matched narrative with its domain. Therefore, an interesting question is whether LLM would perform best in graph recall only if the narrative style of the graphs matches the domain.

To this end, we conduct cross-evaluation over the five different application domains and their corresponding narrative styles as introduced in Sec.3.1. More concretely, for graphs from each domain, we describe it in five different ways, corresponding to the five different domains. The resulting performance is visualized as heatmaps in Fig.3 (a) - (c). Appendix E provides the full table.

**Result Analysis.** The heatmaps of GPT-4 and GPT-3.5 support our conjecture: the diagonals (corresponding to a matching between narrative style and the domain of the data) tend to have higher performance. Such an effect seems to be more prominent with better-performing models.

We find this result striking, that the LLM should do better when the graph is described in the narrative language of the domain that it comes from. While it is an intuitively sensible conjecture that this might help, it is very interesting that this conjecture is borne out so clearly in the results.

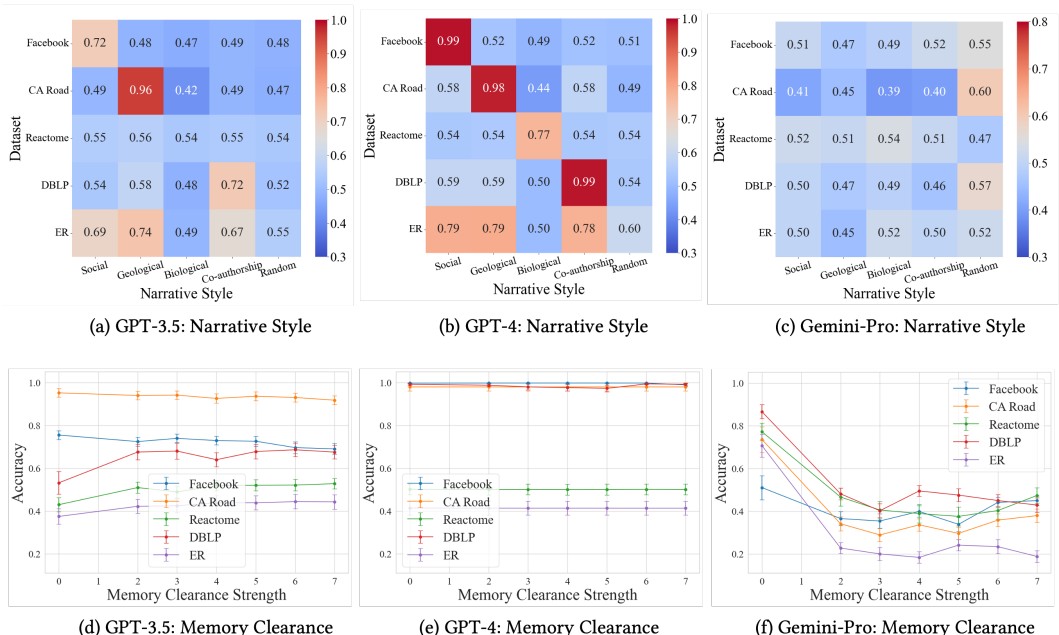

Figure 3: Different factors that influence LLM's graph recall. **(a) - (c): narrative styles.** The heatmaps show that more advanced LLMs like GPT-4 yield best recall accuracy when the graph is narrated in a language style consistent with its original domain. **(d) - (f): memory clearance.** Gemini-Pro appears more sensitive to small noise in context, while GPT's are more robust.

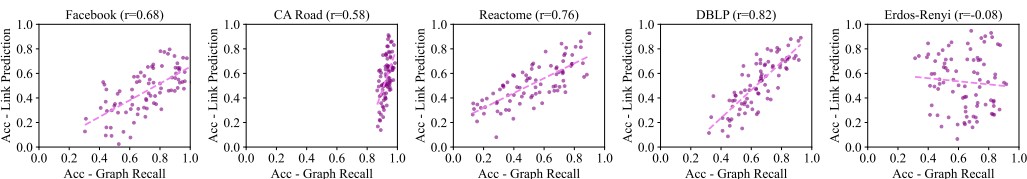

Figure 4: Correlation between GPT-3.5's performance at graph recall ($y$) and link prediction ($x$).

Given this, it is natural to ask whether the result might be coming from the mechanics of the training; in particular, is it possible that the LLM is just reciting text from its training corpus, since the five datasets we use are all public on the Internet? We find this very unlikely, for a simple reason: while the structured graph data comes from the Internet, the narrative descriptions do not; they were generated by us from a simple template for purposes of this experiment, as explained in Sec.3.1.

Since superficial explanations do not seem to explain the strength of the results, it becomes reasonable to suppose that the narrative style is indeed helping with the graph recall task. There are natural, if subtle, reasons why this may indeed be the case: since organically produced text describing road networks, for example, refers a different distribution over graph structures than organically produced text describing social networks, it is a plausible mechanism that recall is helped when the distributional properties in the text align with the distributional properties of the graph that the text describes. An implication is thus that LLMs, especially GPT-4, may have indeed formed a good understanding of the different distributions of graph structures from different domains, as otherwise they wouldn't be able to so consistently perform better when the narrative matches the data domain.

## 4.2 Strength of Memory Clearance

**Motivation.** Our next experiment contains memory clearance (word span test) which is a standard component in previous human graph recall tests. We use memory clearance both to align with these human tests and to simulate the real-world situation where an LLM may not be asked to work on the key data immediately after it receives them.

Table 3: Correlation between LLM's graph recall and link prediction on different microstructures.

| Dataset / Task | Microstructure | | | | |
| --- | --- | --- | --- | --- | --- |
| | *Edge* | *Triangle* | *Star* | *Alt-Triangle* | *Alt-2-Path* |
| **Facebook** | | | | | |
| Graph Recall | -3.70+-5.80 | 1.72+-1.34 ↑ | -0.70+-3.00 | -0.91+-2.25 ↓ | 3.31+-1.77 ↑ |
| Link Prediction | -1.01+-0.69 ↓ | 3.24+-1.05 ↑ | -3.33+-7.39 | 1.21+-0.55 ↑ | 4.98+-3.01 ↑ |
| **CA Road** | | | | | |
| Graph Recall | 0.64+-0.91 | 7.31+-3.49 ↑ | -3.46+-1.77 ↓ | -1.47+-0.92 ↓ | 2.35+-1.29 ↑ |
| Link Prediction | 0.33+-0.58 | 5.10+-3.73 ↑ | -7.00+-3.34 ↓ | 1.64+-0.50 ↑ | 2.86+-2.50 ↑ |
| **Reactome** | | | | | |
| Graph Recall | 18.01+-6.22 ↑ | -0.71+-4.62 | 4.96+-3.81 ↑ | -6.32+-4.23 ↓ | 4.43+-4.93 |
| Link Prediction | -9.59+-3.00 ↓ | 7.71+-4.62 ↑ | -4.99+-4.44 ↓ | 8.32+-2.10 ↑ | 4.54+-4.25 ↑ |
| **DBLP** | | | | | |
| Graph Recall | -8.12+-3.25 ↓ | 1.17+-4.43 | 7.17+-2.44 ↑ | -11.16+-4.35 ↓ | 5.77+-3.76 ↑ |
| Link Prediction | -6.81+-5.44 ↓ | 6.11+-4.68 ↑ | -2.85+-2.47 ↓ | 7.43+-4.04 ↑ | 2.83+-1.72 ↑ |
| **Erdős–Rényi** | | | | | |
| Graph Recall | -1.40+-5.01 | 9.57+-4.89 ↑ | 1.40+-2.71 | -0.41+-4.19 | 3.36+-2.76 ↑ |
| Link Prediction | -2.51+-3.79 | 8.44+-4.26 ↑ | -0.35+-2.27 | 7.59+-6.22 ↑ | 3.38+-2.55 ↑ |

Since both GPT and Gemini are able to perform 100% correctly in the word span test, by design principle we always need to progress to the maximum set of seven sentences. This makes the memory clearance a significant source of noisy context between LLM's first sight of the graph and the final prompt of the recall question. Therefore, it is natural to wonder if the relatively poor performance of LLMs in this graph recall test could have resulted from too much noisy context. In this mini-study, we investigate how different strengths of memory clearance measured would affect the performance.

The strength of memory clearance can be naturally measured by the maximum number of sentences of which the subject proceeds to recite the final words. The number in the standard word span test ranges from 2 to 7, and 0 if the test is dropped. Therefore, we vary this number in our experiment and record the performance of each model on each dataset.

**Result Analysis.** The results are shown in Figure 3 (d) - (f), and we have the following findings.

**LLMs can significantly differ on their sensitivity to small amount of noise in graph recall.** The trends are clear from the three line plots: Gemini-Pro's performance plunges in the first few clearance levels before it touches the bottom. GPT's performance, in contrast, remain more stable, or even increases at initial clearance levels. This indicates that many of our results for GPT models may still likely hold when there is no memory clearance, *i.e.* prompt is given immediately after relevant context – which is default setting of most previous studies.

**Performance of GPT-3.5's and Gemini-pro's is poor even when the question prompt is provided immediately after the relevant context.** This is obvious from the line plots' intersections with y-axis, and perhaps a bit surprising to people who have primarily focused on using LLMs for more challenging graph tasks. This result also helps eliminate the chance that the mediocre performance comes from overly strong memory clearance module that we've installed in place. To boost LLM's ability to reason on graphs, we may need to first figure out how to help them better attend to the correct edge before we seek to improve other more advanced aspects of reasoning. See Sec. 6 for more discussion.

## 5 Correlation between LLM's Graph Recall and Link Prediction

**Motivation.** The graph recall task should not be confused with How do microstructures and performance of LLM's graph recall affect its behavior in other graph reasoning tasks? In this mini-study, we conduct a correlation analysis of LLM's behavior in the label-free link prediction task, which is an important graph reasoning task for LLMs [24]. We primarily experiment with GPT-3.5 because its graph recall exhibits more significant microstructural patterns than GPT-4, and meanwhile have larger performance variation than Gemini-Pro.

**Procedures.** For each graph in the five datasets, we remove 20% of their edges as missing edges. The LLM is then asked to predict those missing edges. Two types of correlation are studied: (1) accuracy

correlation: for each graph in the dataset, we evaluate the LLM's link prediction accuracy ($x$) and graph recall accuracy ($y$) then map these results onto a scatter plot; (2) microstructural correlation: similar to Table 1 and 2, we evaluate and compare the microstructural coefficients of the predicted graph (in link prediction) and the recalled graph (in graph recall test). Table 3 shows the results.

### 5.1   Result Analysis.

Figure 4 shows the scatter plots for accuracy correlation; Table 3 shows microstructural correlation.

**LLM's link prediction performance correlates well with its graph recall performance on real-world graphs**. This is clear from the scatter plot and the $r$ values. For Erdős–Rényi graphs, the correlation is close to zero, which is unsurprising though because the link prediction on random graphs cannot do better than random guess.

**Different tasks may trigger behavior changes of LLMs that can be subtly revealed by their microstructural bias.** Table 3 shows that LLM's microstructural bias in both tasks tend to be positively correlated on triangles and alt-2-paths, and negatively correlated on alt-triangles. We do not have an intuitive explanation for this result. However, this result does indicate that different tasks can trigger certain interesting behavior changes of LLMs that can be subtly revealed by examining their microstructure patterns, which shows the meaningfulness of our study.

## 6   What to Inform about Future Research: an Empirical Perspective

We consider this study as a step towards the long-term agendas both for improving LLM's graph reasoning ability and for further integrating LLM graph analysis into social-science applications. Here we discuss how our findings may translate into actionable bias mitigation strategies and architectural improvements. Appendix F further discusses **limitations** of the work.

- The graph recall test is one of the simplest and most fundamental graph reasoning tasks. Since advanced LLMs yield unsatisfactory performance in this test, we need to re-examine the recent development of approaches that attempt to directly solve more challenging graph reasoning tasks. Meanwhile, notice the association between graph recall test and the "Needle in a Haystack" test (*i.e.* random fact retrieval from long context) [26, 19]: in some sense, the former can viewed as a "graph-customized" version of the latter. Therefore, existing techniques for boosting LLM's performance in the "Needle in a Haystack" test, e.g. recurrent memory augmentations [26], may likely benefit LLM's graph recall ability as well.

- Our experiment shows that LLMs tend to favor more triangles and alternating 2-paths. Researchers of Graph Foundation Models (GFM) [31, 33, 59, 29] should be alerted of such systematic bias. Because the procedures for training GFMs are largely inspired by those for LLMs, similar retrieval bias could emerge. A potential mitigation strategy worthy of exploration is to consider balancing/compensating the frequencies of different graph motifs that occur in the training data.

- We have also found that more advanced LLM's performance have a striking dependence on the compatibility between the application domain of a real-world graph and its narrative style. This hints a potential direction for designing more powerful graph encoding for LLMs via certain domain adaptation strategies – by combining the adaptation strategy with either the graph-to-text methods [18, 20] or the graph-to-embedding methods [12, 37, 13].

## 7   Related Work

### 7.1   Humans in the Graph Recall Task

There have been decades of work studying human's graph recall ability, primarily focused on their memory and recall of social networks. This line of work originates from sociologist's need to collect real-world social networks by asking people to recall their social relationships. [40] noticed that people forget a significant portion of their social networks, so they built a to predict missing links from recall. [9] more closely study the mechanism of the forgetting of social networks. [5] found that humans can more accurately recall social networks that contain more certain microstructural patterns such as triangles or cliques.

Following the previous works, [8] establishes experimental protocols for examining the performance and microstructural patterns of human's recall of social networks. In their study, each subject reads a short, artificial description of the relationships among a number of 15 people: "Henry is a member of the same club as Elizabeth. James sings in a choir with Anne ...". The subject is then asked to name who has interacted with whom in the experiment. The authors are also among the first to verify that humans' graph recall tend to exhibit patterns of triadic closure. [7] further investigates how sex affects human's recall. [6] uses a signed network model to show that human's graph recall may repel certain unstable patterns that involve unbalanced relationships between enemies and friends. Since LLMs are trained on human-readable corpus, many ideas and findings on human's graph recall are instructional to our exploration of LLM's graph reasoning ability. Further studies show that a person's ability to accurately recall a social network has profound influence on their social decisions [25, 11, 52, 53, 54].

The graph recall test is a meaningful test for both humans and LLMs, though their experiment outcomes may need to be interpreted in a subtly different way. The graph recall test is meaningful for LLMs because we observe their performance to be far from perfect in the test. Compared with humans, however, LLMs may face different challenges. Human's bottleneck in this test is their limited brain capacity, while LLMs may have difficulty in always attending to the correct positions in distant earlier context (or in precisely encoding context into hidden states for RNN-based models).

### 7.2 Graph Reasoning with Large Language Models

Graph reasoning with LLMs is an active research area in recent years. On the benchmark and methodology level, datasets and frameworks are developed to integrate graphs with LLMs [49, 36]. In addition, researchers are exploring ways to solve question answers over structured data with LLMs in a unified way [23]. Others are also trying to advance the prompting capabilities of LLMs by mimicking the connective manner of human thinking or elaboration[4]. There has also been work on developing general graph models to handle various graph tasks with a unified framework [30]. More recently, [43] provides theoretical analysis on the limit of transformer's reasoning capability on graphs, by relating transformer's capabilities on graph reasoning to the computational complexity of related tasks. On the application level, LLMs have been adopted in graph reasoning applications such as node classification [21, 14], graph classification [38, 60], knowledge graphs [58, 56], and recommendation systems [55]. We are still lacking a data perspective to address the basic question of whether LLMs can accurately remember the graph that they are supposed to reason upon, which is a prerequisite for any advanced graph tasks. Inspired and supported by cognitive studies, we conduct the first comprehensive investigation of graph recall by LLMs, filling a gap in the existing literature.

## 8 Conclusion

This work proposes and studies graph recall as a simple yet fundamental task for understanding LLM's graph reasoning abilities. We design and conduct systematical studies on the accuracy and biased microstructures of LLM's graph recall, by creatively drawing its connection with the existing cognitive studies on humans. Future work may examine more varieties of microstructural patterns including higher-order structures [51] and "sense of distance", both of which which are crucial for understanding graph structures [28, 50, 57]. Another direction is to study how to improve LLM's graph recall by prompting or other methods.

## Acknowledgment

We thank Matt Brashears, Eric Quintane for their help in discussing their earlier work on network recall and its relation to the current project, as well as their help with data relevant to their work. We also thank Tianxiang Zhao for his insights into the intersection of graphs and LLMs. Their generous assistance was a great benefit to our project. Our work here has been supported in part by a Vannevar Bush Faculty Fellowship, AFOSR grant FA9550-19-1-0183, a Simons Collaboration grant, a grant from the MacArthur Foundation, and a grant from the Microsoft AFMR program.

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

# Appendix

## A  Code and data

Our code and data can be downloaded at: `https://github.com/Abel0828/llm-graph-recall`.

## B  More Experimental Details

**Dataset.**  For the experiment in Sec.3.1 and 3.2, Fig.5-7 show graph samples of the Facebook, Reactome, and Erdos-Renyi datasets. The graph narratives are shown in Appendix B.1. The sentence sets for memory clearance are shown in Appendix B.2.1.

For the experiment in Sec.3.3, we utilize the two social networks in [8]'s human experiment. Each social network was presented to the test taker in two different narrative styles: one is friendship-based, and the other is kinship-based *e.g.* "James is the brother of Anne...". In Brashears' experiment, a test taker has equal chance to see one of the two narrative style (but never both). The final result in their paper, however, was reported in an aggregated form that mixes up response to both narrative styles, as the authors reported to see no difference between caused by these two. To stay aligned we choose do the same with LLMs, but make a note that LLMs may behave differently to different graph narrative styles, which we have further investigated in a separate study, Sec.4.1.

We need to be careful with the generation of the node names, because it affects how much parametric knowledge in the LLM gets elicited to aid the graph recall process. We general guideline is that, although the parametric knowledge can potentially create a shortcut for LLM's handling of the graph recall task, this should not be forcefully forbidden as it is also seen in real-world applications. Therefore, we have two cases when creating our datasets, namely (1) where parametric knowledge is potentially useful, and (2) where parametric knowledge is less useful.

- For protein networks, the protein names are not random. They are unique protein identifiers (known as the "UniProtKB/Swiss-Prot accession number" or NCBI index). Each node is assigned its real name. We also confirmed that LLMs know and precisely understand those protein identifiers. The same is true for DBLP coauthorship networks.

- For all other networks, the node names are generated and randomly assigned. For example, in traffic (geographical) networks, the node names are "bank", "townhall", "high school", etc.

**Computing Resources.**  We use the Azure OpenAI service to test GPT-based models. For Gemini-Pro, we use the `gemini-pro` model API provided by Google's Generative AI. For Llama Family models, we use the open-sourced models `meta-llama/Llama-2-7b-hf` and `meta-llama/Llama-2-13b-hf` on Hugging Face, tuned on two Quadro RTX 8000 GPUs with 48 GB of RAM.

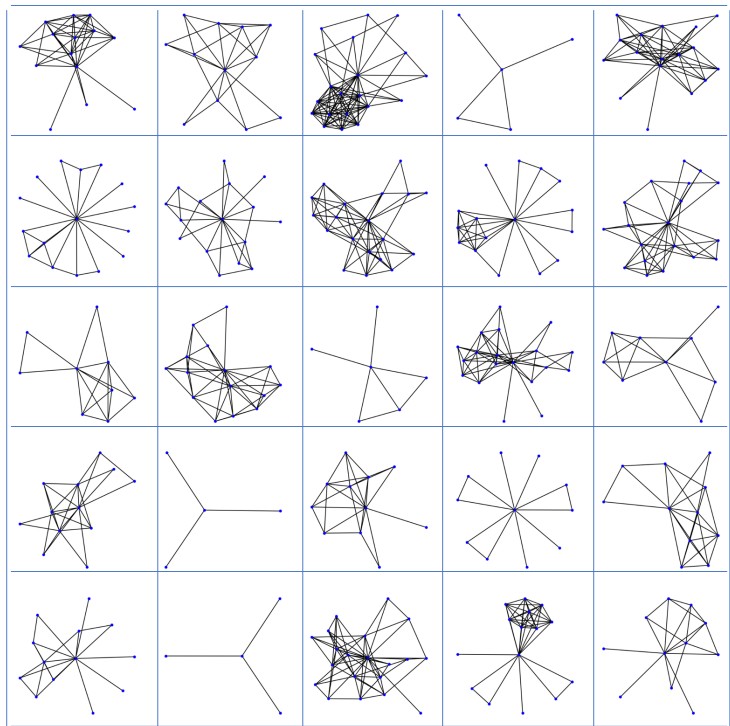

Figure 5: Graph samples of the Facebook dataset.

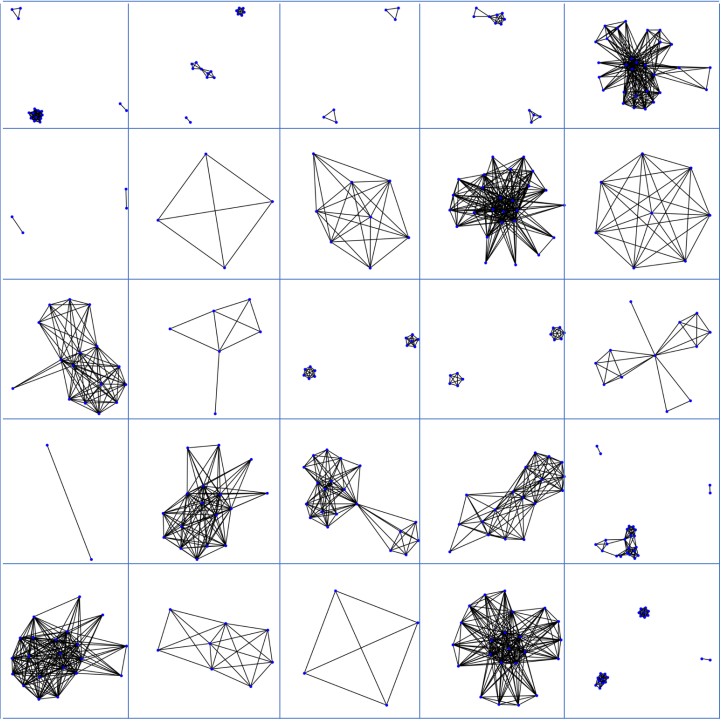

Figure 6: Graph samples from the Reactome (protein-protein interaction) dataset.

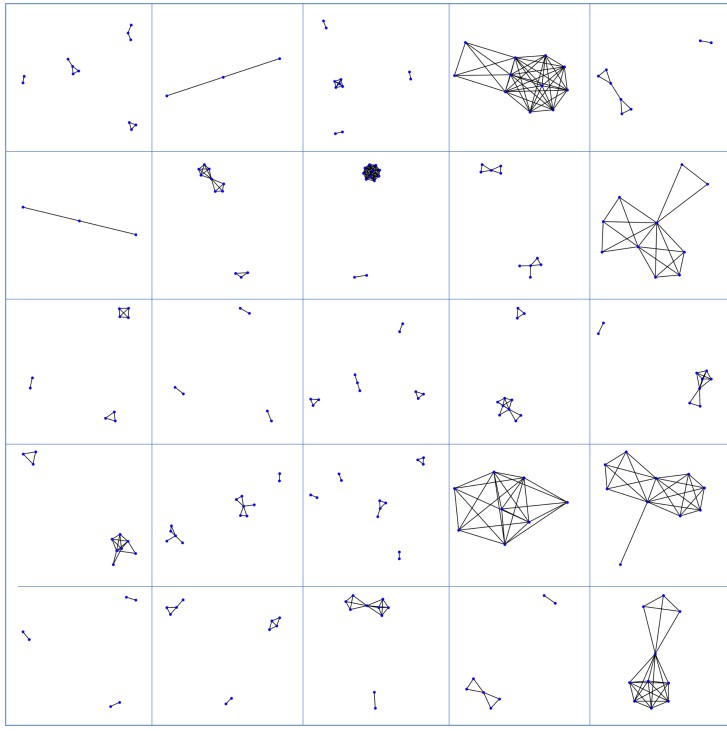

Figure 7: Graph samples of the Erdos-Renyi dataset.

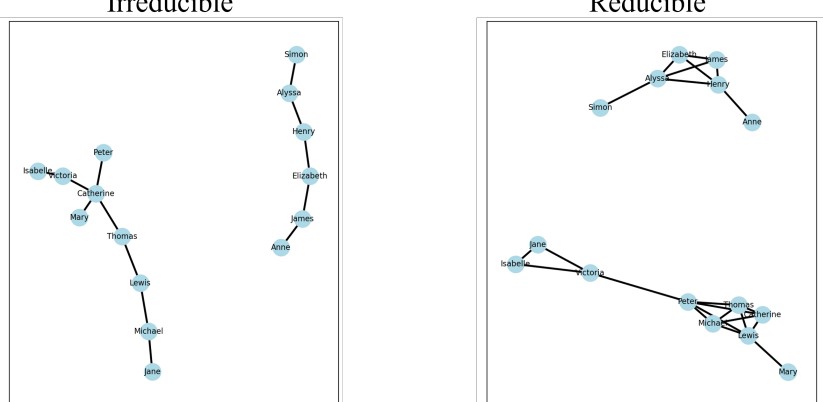

Figure 8: The two 15-node social networks with different connectivity patterns used in [8] and in our Sec. 3.3 experiment.

## B.1 Samples of the Graph Narratives for All Datasets

**Facebook**

"The following people have friendship relations: Harper, James, Abigail, Noah, Alexander, Oliver, William, Charlotte, Sophia, Benjamin, Emily, Daniel, Ethan, Henry. Their have friendships: Harper and Daniel are friends. Harper and Sophia are friends. James and William are friends. James and Daniel are friends. James and Abigail are friends. James and Sophia are friends. James and Noah are friends. Abigail and Benjamin are friends. Abigail and Noah are friends. Abigail and Ethan are friends. Abigail and William are friends. Noah and Charlotte are friends. Noah and Benjamin are friends. Noah and Emily are friends. "

**Traffic Network**

We have a traffic network that involves the following destinations: Bank, Town Hall, Grocery Store, High School. The traffic network is the following. The traffic can directly flow from the Bank to the Grocery Store. The traffic can directly flow from the Town Hall to the Grocery Store. The traffic can directly flow from the Grocery Store to the High School.

**Protein Interaction**

The following proteins have mutual interactions: Q13563, Q16280, P61006, O75385, Q9ULV0, O75154, Q7L804, Q15907, Q96QF0, Q8IV77, P08100, P18085, Q14028, P98161, Q9ULH1, Q9P2M4. Their interactions are: Protein Q13563 interacts with Protein P98161. Protein Q13563 interacts with Protein P08100. Protein Q13563 interacts with Protein Q8IV77. Protein Q13563 interacts with Protein Q16280. Protein Q13563 interacts with Protein Q14028. Protein Q13563 interacts with Protein P18085. Protein Q13563 interacts with Protein Q9ULH1. Protein Q13563 interacts with Protein O75154. Protein Q13563 interacts with Protein P61006. Protein Q13563 interacts with Protein Q96QF0. Protein Q16280 interacts with Protein P98161. Protein Q16280 interacts with Protein P08100. Protein Q16280 interacts with Protein Q8IV77. Protein Q16280 interacts with Protein Q14028. Protein Q16280 interacts with Protein P18085. Protein Q16280 interacts with Protein Q9ULH1. Protein Q16280 interacts with Protein O75154. Protein Q16280 interacts with Protein P61006. Protein Q16280 interacts with Protein Q96QF0. Protein P61006 interacts with Protein Q96QF0. Protein P61006 interacts with Protein P98161. Protein P61006 interacts with Protein P08100. Protein P61006 interacts with Protein Q8IV77. Protein P61006 interacts with Protein Q14028. Protein P61006 interacts with Protein Q9ULH1. Protein P61006 interacts with Protein O75154. Protein O75385 interacts with Protein Q15907. Protein O75385 interacts with Protein Q9P2M4.

**Erdos-Renyi Graph**

A graph has the following nodes: Node 0, Node 1, Node 2, Node 3, Node 4, Node 5, Node 6, Node 7, Node 8, Node 9, Node 10, Node 11, Node 12, Node 13, Node 14, Node 15, Node 16, and the following edges: Node 0 is connected with Node 1. Node 0 is connected with Node 2. Node 0 is connected with Node 3. Node 0 is connected with Node 4. Node 0 is connected with Node 6. Node 0 is connected with Node 7. Node 0 is connected with Node 8. Node 0 is connected with Node 10. Node 0 is connected with Node 11. Node 0 is connected with Node 12. Node 0 is connected with Node 13. Node 0 is connected with Node 14. Node 1 is connected with Node 4. Node 1 is connected with Node 5. Node 1 is connected with Node 7. Node 1 is connected with Node 8. Node 1 is connected with Node 9. Node 1 is connected with Node 10. Node 1 is connected with Node 11. Node 1 is connected with Node 14. Node 1 is connected with Node 16. Node 2 is connected with Node 4. Node 2 is connected with Node 5. Node 2 is connected with Node 6. Node 2 is connected with Node 7. Node 2 is connected with Node 9. Node 2 is connected with Node 10. Node 2 is connected with Node 11. Node 2 is connected with Node 13. Node 2 is connected with Node 14. Node 2 is connected with Node 16.

## B.2 Textual Samples Used in Experiment

### B.2.1 Sentence Sets for Memory Clearance

**Two sentence sets**

"The ghillie suit, the modern sniper's principle camouflage uniform, derives its name from Scottish game hunters."

"Industrial accidents- explosions of stored oil and gas- are bad enough on land."

**Three sentence sets**

"But overall, the teenage share of the population wasn't getting much bigger."

"This is a system steeped in tradition, and I think that's part of the problem."

"We're unable to move, our legs stuck beneath us as under a great weight."

...

**Seven Sentence Set**

"One afternoon I open a letter from my younger sister, the photo chronicler of family events."

"Although they have enough food to sustain your group for years, supermarkets are also dangerous."

"Maternity, or additional offspring, may force upon the woman a distressful life and future."

"By Wednesday night's vote meeting, Sabrina was thoroughly disgusted by the superficiality of the week."

"I ask him to please stop lying about trying to take my place in the war."

"Since the 1950's, freeways have been built through every large and medium-sized city."

"Instead, he took a job in Washington, analyzing weapons expenditures for the U.S. Navy."

# C Full Comparison of True Graph and LLM Recall Graph

We provide a full comparison of the true graph and the recall graph from different LLMs. The results are shown in Table 4, Table 5, and Table 6, respectively.

Table 4: Full comparison of true graph and GPT-3.5 recall graph on different microstructures.

| Dataset | Edge | Triangle | Star | Alt-Triangle | Alt-2-Path |
|---|---|---|---|---|---|
| | | | Microstructure | | |
| **Facebook** | | | | | |
| True Graph | 0.88+-1.46 | 0.61+-0.57 | -1.75+-0.45 | -0.02+-0.24 | 1.07+-0.21 |
| Recall Graph | -2.82+-5.47 | 2.33+-1.33 | -2.46+-3.00 | -0.93+-2.23 | 4.38+-1.72 |
| Diff | -3.70+-5.80 | 1.72+-1.34 | -0.70+-3.00 | -0.91+-2.25 | 3.31+-1.77 |
| **CA Road** | | | | | |
| True Graph | 0.86+-2.50 | 2.90+-1.79 | -4.83+-1.53 | -2.07+-0.37 | 4.75+-0.52 |
| Recall Graph | 1.51+-1.73 | 10.21+-1.74 | -8.29+-2.10 | -3.54+-1.05 | 7.10+-0.91 |
| Diff | 0.64+-0.91 | 7.31+-3.49 | -3.46+-1.77 | -1.47+-0.92 | 2.35+-1.29 |
| **Reactome** | | | | | |
| True Graph | 18.19+-5.96 | 1.23+-4.31 | -9.46+-2.50 | 1.15+-2.61 | 1.40+-3.28 |
| Recall Graph | 0.18+-5.92 | 0.52+-2.78 | -4.49+-5.15 | -5.17+-3.49 | 5.84+-4.11 |
| Diff | -18.01+-6.22 | -0.71+-4.62 | 4.96+-3.81 | -6.32+-4.23 | 4.43+-4.93 |
| **DBLP** | | | | | |
| True Graph | 7.63+-3.24 | -1.68+-2.02 | -5.30+-2.41 | 2.69+-2.14 | -1.17+-2.30 |
| Recall Graph | -0.49+-2.24 | -0.51+-3.86 | 1.88+-1.79 | -8.47+-4.16 | 4.60+-3.58 |
| Diff | -8.12+-3.25 | 1.17+-4.43 | 7.17+-2.44 | -11.16+-4.35 | 5.77+-3.76 |
| **Erdős–Rényi** | | | | | |
| True Graph | -0.19+-3.44 | -1.96+-1.59 | -0.75+-3.28 | 0.86+-1.66 | 1.38+-0.89 |
| Recall Graph | -1.60+-5.10 | 7.61+-4.75 | 0.64+-1.66 | 0.45+-3.50 | 4.74+-2.53 |
| Diff | -1.40+-5.01 | 9.57+-4.89 | 1.40+-2.71 | -0.41+-4.19 | 3.36+-2.76 |

Table 5: Full comparison of true graph and GPT-4 recall graph on different microstructures.

| Dataset | Edge | Triangle | Star | Alt-Triangle | Alt-2-Path |
|---|---|---|---|---|---|
| | | | Microstructure | | |
| **Facebook** | | | | | |
| True Graph | 0.88+-1.46 | 0.61+-0.57 | -1.75+-0.45 | -0.02+-0.24 | 1.07+-0.21 |
| Recall Graph | 0.71+-1.51 | 0.66+-0.61 | -1.69+-0.45 | -0.03+-0.26 | 1.08+-0.21 |
| Diff | -0.17+-0.50 | 0.05+-0.78 | 0.06+-0.21 | -0.01+-0.26 | 0.01+-0.06 |
| **CA Road** | | | | | |
| True Graph | 0.86+-2.50 | 2.90+-1.79 | -4.83+-1.53 | -2.07+-0.37 | 4.75+-0.52 |
| Recall Graph | 2.20+-1.55 | 8.97+-2.43 | -7.93+-1.52 | -3.34+-1.15 | 6.60+-0.78 |
| Diff | 1.34+-1.62 | 6.07+-3.93 | -3.09+-2.08 | -1.27+-0.96 | 1.85+-0.91 |
| **Reactome** | | | | | |
| True Graph | 18.19+-5.96 | 1.23+-4.31 | -9.46+-2.50 | 1.15+-2.61 | 1.40+-3.28 |
| Recall Graph | 6.64+-5.56 | 2.05+-2.31 | -6.77+-3.21 | -0.84+-1.24 | 0.95+-0.97 |
| Diff | -11.54+-5.82 | 0.82+-4.28 | 2.69+-2.87 | -1.99+-2.03 | -0.46+-2.14 |
| **DBLP** | | | | | |
| True Graph | 7.63+-3.24 | -1.68+-2.02 | -5.30+-2.41 | 2.69+-2.14 | -1.17+-2.30 |
| Recall Graph | 6.37+-3.08 | -3.09+-3.33 | -5.99+-3.24 | 4.40+-3.10 | -2.76+-2.51 |
| Diff | -1.26+-2.56 | -1.41+-3.26 | -0.69+-3.74 | 1.71+-2.96 | -1.59+-2.24 |
| **Erdős–Rényi** | | | | | |
| True Graph | -0.19+-3.44 | -1.96+-1.59 | -0.75+-3.28 | 0.86+-1.66 | 1.38+-0.89 |
| Recall Graph | -1.69+-3.21 | -0.01+-1.09 | -0.23+-2.13 | -0.40+-1.49 | 0.93+-1.76 |
| Diff | -1.49+-4.15 | 1.95+-1.54 | 0.52+-2.79 | -1.26+-1.66 | -0.45+-0.76 |

Table 6: Full comparison of true graph and Gemini-Pro recall graph on different microstructures.

| Dataset | Edge | Triangle | Star | Alt-Triangle | Alt-2-Path |
|---|---|---|---|---|---|
| | | | Microstructure | | |
| **Facebook** | | | | | |
| True Graph | 0.88+-1.46 | 0.61+-0.57 | -1.75+-0.45 | -0.02+-0.24 | 1.07+-0.21 |
| Recall Graph | -1.44+-0.90 | -1.68+-3.00 | -0.25+-2.35 | 2.38+-1.35 | 1.74+-1.06 |
| Diff | -2.31+-1.26 | -2.29+-2.99 | 1.50+-2.37 | 2.40+-1.37 | 0.67+-1.10 |
| **CA Road** | | | | | |
| True Graph | 0.86+-2.50 | 2.90+-1.79 | -4.83+-1.53 | -2.07+-0.37 | 4.75+-0.52 |
| Recall Graph | 1.70+-1.15 | 4.93+-0.84 | 0.41+-2.95 | -1.18+-0.44 | -2.07+-2.28 |
| Diff | 0.84+-1.68 | 2.02+-0.29 | 5.24+-0.27 | 0.89+-0.28 | -6.82+-0.62 |
| **Reactome** | | | | | |
| True Graph | 18.19+-5.96 | 1.23+-4.31 | -9.46+-2.50 | 1.15+-2.61 | 1.40+-3.28 |
| Recall Graph | 6.25+-5.81 | 2.50+-4.04 | 4.02+-5.87 | 4.47+-5.71 | 5.56+-5.01 |
| Diff | -11.94+-4.65 | 1.27+-5.22 | 13.47+-4.70 | 3.32+-4.57 | 4.15+-4.01 |
| **DBLP** | | | | | |
| True Graph | 7.63+-3.24 | -1.68+-2.02 | -5.30+-2.41 | 2.69+-2.14 | -1.17+-2.30 |
| Recall Graph | -11.84+-4.72 | -3.40+-3.83 | 5.94+-2.13 | -8.39+-3.34 | 9.66+-3.99 |
| Diff | -19.47+-3.30 | -1.72+-3.45 | 11.24+-2.45 | -11.08+-3.90 | 10.83+-4.53 |
| **Erdős–Rényi** | | | | | |
| True Graph | -0.19+-3.44 | -1.96+-1.59 | -0.75+-3.28 | 0.86+-1.66 | 1.38+-0.89 |
| Recall Graph | -3.06+-1.67 | -0.45+-4.00 | -1.41+-1.20 | 1.45+-2.17 | 0.69+-0.46 |
| Diff | -2.86+-5.16 | 1.51+-5.04 | -0.66+-3.95 | 0.59+-2.81 | -0.70+-1.15 |

# D  Sex Priming as a Mini-study

**Motivation.** [7] interestingly found that females can more accurately recall their social networks than males. The underlying explanation is that underrepresented groups tend to be more aware of their surroundings. We conjecture that this trend might also exist in the corpus on which LLMs are trained, and therefore wonder whether LLMs perform better at graph recall when asked to play the role of a female.

**Procedures.** Aligned with [7]'s design, we include at the beginning of the test (i.e. prior to all steps) a sex prime, which is a piece of text designed to elicit the test subject's awareness of their sex. [7]'s sex prime is a short question that asks the subject's opinion on same-sex versus mixed-sex housing. However, this method fails with LLMs because they refuse to identify as having any pre-given sex. Instead, we send role-playing instructions to LLMs by stating their "sex" in the system message at the beginning of the chat. We further confirm the successful elicitation by asking what their sex is afterwards. We compare the LLM's graph recall performance under male and female roles.

**Result Analysis.** The results are shown in Fig.9 (g) - (i). The error bars are 95% confidence intervals. The effect of sex roles appear to be insignificant in most cases, which negates our initial conjecture and interestingly opposes the existing findings on humans. In fact, LLMs underperform in both roles when compared with the control group, *i.e.* cross-referencing Table 1 "Accuracy" column.

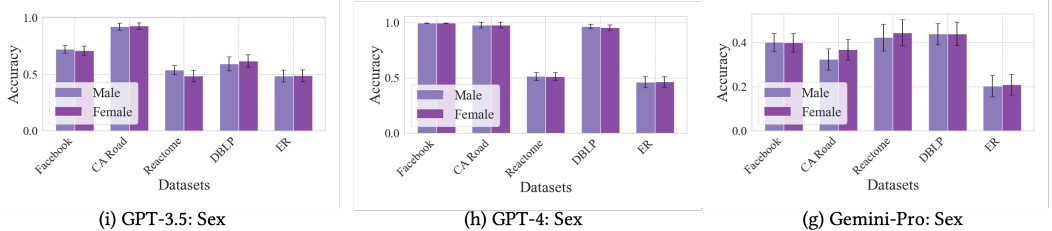

(i) GPT-3.5: Sex        (h) GPT-4: Sex        (g) Gemini-Pro: Sex

Figure 9: Different factors that influence LLM's graph recall. **(g) - (i): sex.** Following [7]'s procedures, we test and find that the effect of sex roles that an LLM is asked to play is insignificant on their capability of graph recall — a result different from that on humans.

## E  Full Performance Metrics with Different Narrative Styles

The full performance comparison of different narrative styles on five datasets for each LLM is present in Table 7.

Table 7: Full graph recall performance metrics with different narrative styles on five datasets.

| Dataset | Narrative Style | GPT-3.5 (%) | | | | GPT-4 (%) | | | | Gemini-Pro (%) | | | |
|---|---|---|---|---|---|---|---|---|---|---|---|---|---|
| | | F1 | Accuracy | Precision | Recall | F1 | Accuracy | Precision | Recall | F1 | Accuracy | Precision | Recall |
| Facebook | Social | 72.34 | 71.60 | 66.84 | 90.22 | 99.75 | 99.80 | 99.51 | 100.00 | 34.56 | 51.13 | 34.60 | 49.11 |
| | Geological | 44.39 | 48.35 | 38.07 | 55.89 | 37.55 | 52.18 | 37.50 | 37.61 | 41.42 | 46.71 | 35.27 | 66.10 |
| | Biological | 51.99 | 47.11 | 40.48 | 75.91 | 49.93 | 48.64 | 39.95 | 67.72 | 31.01 | 48.79 | 27.67 | 49.88 |
| | Co-authorship | 42.30 | 48.76 | 38.25 | 53.69 | 37.98 | 51.87 | 37.42 | 38.59 | 28.95 | 51.84 | 27.08 | 42.27 |
| | Random | 47.57 | 47.81 | 40.28 | 63.03 | 44.57 | 51.27 | 40.41 | 51.54 | 24.36 | 55.25 | 31.39 | 27.30 |
| CA Road | Social | 41.37 | 49.30 | 36.43 | 53.07 | 40.85 | 58.22 | 40.85 | 40.85 | 28.19 | 40.53 | 25.78 | 45.17 |
| | Geological | 92.95 | 95.52 | 89.81 | 97.13 | 98.00 | 98.11 | 98.00 | 98.00 | 31.92 | 44.69 | 28.02 | 53.02 |
| | Biological | 44.90 | 41.59 | 34.74 | 71.64 | 44.08 | 43.73 | 34.72 | 66.25 | 18.74 | 38.60 | 14.20 | 40.75 |
| | Co-authorship | 40.65 | 48.71 | 35.75 | 53.66 | 40.50 | 57.91 | 40.38 | 40.67 | 24.08 | 40.15 | 18.40 | 46.41 |
| | Random | 31.88 | 47.26 | 29.39 | 38.14 | 31.43 | 49.10 | 29.38 | 35.58 | 31.52 | 59.93 | 33.23 | 35.46 |
| Reactome | Social | 56.32 | 54.99 | 49.54 | 71.59 | 50.68 | 53.69 | 50.45 | 50.91 | 43.34 | 51.58 | 47.48 | 52.57 |
| | Geological | 57.47 | 56.18 | 51.41 | 68.15 | 50.09 | 54.05 | 50.41 | 50.06 | 48.72 | 51.28 | 44.97 | 63.11 |
| | Biological | 44.47 | 53.68 | 46.53 | 54.62 | 76.80 | 77.04 | 77.57 | 77.44 | 46.59 | 54.09 | 49.06 | 55.07 |
| | Co-authorship | 50.10 | 54.55 | 51.48 | 54.95 | 50.21 | 53.72 | 50.42 | 50.12 | 31.68 | 51.29 | 36.57 | 36.24 |
| | Random | 56.27 | 53.67 | 52.77 | 64.92 | 53.59 | 54.11 | 53.13 | 56.12 | 24.92 | 46.62 | 43.65 | 26.30 |
| DBLP | Social | 51.00 | 53.79 | 42.65 | 68.22 | 42.05 | 58.79 | 42.05 | 42.05 | 35.72 | 49.70 | 36.09 | 51.83 |
| | Geological | 47.01 | 58.17 | 43.30 | 53.69 | 42.10 | 58.81 | 42.09 | 42.11 | 41.89 | 47.02 | 36.96 | 66.93 |
| | Biological | 51.75 | 48.30 | 41.44 | 75.21 | 50.60 | 50.30 | 41.67 | 69.24 | 41.95 | 49.15 | 33.76 | 69.52 |
| | Co-authorship | 65.08 | 72.47 | 67.56 | 74.97 | 97.88 | 98.70 | 97.17 | 98.81 | 47.36 | 46.33 | 40.23 | 77.56 |
| | Random | 45.22 | 52.35 | 40.08 | 54.86 | 42.56 | 53.58 | 39.98 | 47.07 | 15.82 | 57.50 | 32.94 | 13.06 |
| ER | Social | 61.69 | 68.95 | 51.67 | 81.15 | 62.60 | 78.68 | 62.40 | 62.80 | 39.37 | 50.41 | 44.01 | 56.08 |
| | Geological | 62.70 | 73.89 | 56.07 | 73.67 | 64.62 | 79.13 | 64.37 | 64.89 | 43.18 | 44.92 | 41.96 | 71.69 |
| | Biological | 47.84 | 49.09 | 39.65 | 69.47 | 46.11 | 50.42 | 39.69 | 62.71 | 36.56 | 52.32 | 33.54 | 54.60 |
| | Co-authorship | 52.69 | 66.74 | 48.15 | 66.90 | 61.13 | 78.10 | 60.33 | 62.06 | 29.69 | 50.42 | 38.57 | 42.52 |
| | Random | 49.49 | 55.21 | 45.33 | 63.94 | 44.03 | 60.34 | 44.97 | 44.40 | 22.27 | 52.40 | 36.95 | 26.33 |

## F  Limitations

Our evaluation of LLM's graph recall performance is not exhaustive. Due to budget constraint, we only test on datasets from five domains and on a limited number of graph samples from each domain. Also, while we have experimented with several main-stream LLMs including GPT-3.5, GPT-4, Gemini-Pro, and Llama-2, there are other popular LLMs such as Claude 3 and Llama-3 which our experiment have not covered.

Another limitation of our work, which we also consider to be an important direction for future work, lies in the distinction between graph recall and graph retrieval. The latter requires a more variational setting on the amount of contextual noise in graph narratives. In our work, we have intentionally designed the graph narratives to be simple and fixed, because we already observed considerable errors in LLM's graph recall at this starting point (and we know little about LLM's behaviors even in this simplest setting). However, the real-world structure-rich text that LLMs process may contain significantly more contextual noise, where some of the findings in this work may not be easily generalizable (although we conjecture that many systematic bias we have observed will persist). How to both realistically and comprehensively vary the amount of noise in graphs narratives remains to be a challenging topic to explore.

Finally, despite being small, the gap between our result and the recent "Needles in a Haystack" by Greg Kamradt [19] on GPT-4 needs more investigation. In [19], it is reported that GPT-4 makes little errors in long-context recall over Paul Graham essays, up to the context length of 73k. While our study also shows (in Table 1) that GPT-4 performs well on three out of the five datasets used, its performance on the rest two is mediocre. There can be many possible explanations to reconcile the gap, but the lack of robustness that we observe in tests of this kind is an important starting point for further investigation.

## G  Relationship with the "Edge Existence" Task in (Fatemi et al.) [18]

The edge existence task is one of the six exemplary tasks proposed in [18] for measuring LLM's graph reasoning ability. Both our graph recall test and the edge existence task require the LLM to

decide the existence of graph edges described in earlier text. However, our work has the following main differences and novelty.

- The main topics of investigation are different. The central topics in this paper are 1) to unveil biased subgraph patterns (motifs) in LLM's graph recall, (2) to rigorously compare LLM's performance with humans, and (3) to investigate how results of (1) are affected by various factors. These topics are not studied in [18].

- The empirical findings are completely non-overlapped. Our work presents many findings about LLM's biased graph recall patterns and their rigorous comparison with humans, which are novel to [18].

- Our experimental pipelines are different. Our paper uses a more rigorous evaluation pipeline inspired by classical human cognition studies. In addition to synthetic data, we also extensively use real-world datasets for experiment, which was not reported [18].

