# OpenReview forum: "Microstructures and Accuracy of Graph Recall by Large Language Models"
_NeurIPS.cc/2024/Conference — NeurIPS 2024 poster_

### Official Review · Reviewer_QYAY · 2024-07-05

**Soundness:** 3
**Presentation:** 4
**Contribution:** 3
**Rating:** 7
**Confidence:** 4

**Summary:**

This paper conducts a comprehensive evaluation of LLMs' capability to recall graph (sub)structures when using natural languages as the interface. Through extensive experiments on diverse graphs from different domains, it points out interesting phenomena like LLMs' underperformance in graph recall tasks. It sheds light on novel insights and inspires further research in relevant fields.

**Strengths:**

1. This paper is well-written and well-motivated. The idea of integrating computational social science and LLMs, especially comparing the behavior of both LLMs and human beings, is interesting.
2. This paper tackles an important problem. Despite various papers on evaluating the effectiveness of LLMs' reasoning capability for graphs, few evaluate the first step, such as LLMs' capability to accurately memorize graph structures. This paper strengthens this point and makes a valuable contribution. In section 5, the authors also show that LLMs' capability on structure-related tasks is highly correlated to their capability to recall substructures.
3. The evaluation is well-designed, with support from social science.
4. Code implementation is provided, ensuring the reproducibility of experiments.
5. Several potential directions are provided to inspire future research.

**Weaknesses:**

1. This paper primarily studies the case where natural language is used as the interface to represent graphs. However, many recent works [1, 2] demonstrate that using a graph encoder and multi-modal projector may be a better choice to inject graph-related knowledge into LLMs. I wonder whether we would arrive at different conclusions when using such graph language models.

2. The authors focus on the case with k=5 (line 101) in this paper, which somewhat limits the generalizability of the conclusions. I think LLMs' recalling capability may also be closely related to the length of inputs. As a result, those LLMs designed with long-context capabilities may perform better for larger graphs. Moreover, larger graphs may present more complicated substructure patterns and lead to somewhat different results.

3. I cannot find the formulas used to obtain the results for Table 1.

4. Some phenomena in this paper do not have corresponding explanations (line 327).

[1] Perozzi, B., Fatemi, B., Zelle, D., Tsitsulin, A., Kazemi, M., Al-Rfou, R., & Halcrow, J. (2024). Let Your Graph Do the Talking: Encoding Structured Data for LLMs. ArXiv, abs/2402.05862.

[2] Chen, R., Zhao, T., Jaiswal, A., Shah, N., & Wang, Z. (2024). LLaGA: Large Language and Graph Assistant. ArXiv, abs/2402.08170.

**Questions:**

1. Will the conclusions be consistent when we use a multi-modal LLMs and using tokens encoded by graph models like GNNs?
2. How will the conclusion change when we vary the size of the graphs?
3. How to calculate the bias score in Table 1?
4. May you provide more explanations for some phenomena, like one in line 327?
5. There's one recent paper [1] which provides theoretical analysis on transformer's reasoning capability on graphs. They relate transformer's capabilities on graph reasoning to the computational complexity of related tasks. In this case, I think graph recalling (edge level) is also a retrival task, and it would be interestring to discuss the relationships between findings in these two papers.

[1] Sanford, C., Fatemi, B., Hall, E., Tsitsulin, A., Kazemi, S.M., Halcrow, J., Perozzi, B., & Mirrokni, V.S. (2024). Understanding Transformer Reasoning Capabilities via Graph Algorithms. ArXiv, abs/2405.18512.

**Limitations:**

Authors have dicussed the limitations and potential social impacts.

---

> ### Author Rebuttal · Authors · 2024-08-07
>
> We thank the reviewer for the review!
>
> **Question 1**
>
> We believe the conclusions will be relatively close to those of our results obtained when using domain-free narrative style to describe the graph (i.e. naming nodes as "node 1", "node 2", etc.).  Our work has used this narrative style to test “Erdos-Renyi” graphs in Table 1, and to test topology structures extracted from other graph datasets in Figure 3 (a) – (c) “random” columns.
>
>
> This is because the graph encoder [1,2] projects all nodes/edges to the same token space of LLMs, while leaving out all domain-relevant information. This is similar to, though not exactly the same as, one of our settings where the narrative style is domain-free, in which case the node tokens are simply extracted by dictionary lookup.
>
>
> This work still makes meaningful contributions in this context, for two reasons:
>
> - In terms of the proposed graph recall task and evaluation pipeline, they can be directly used to evaluate the “LLM+graph encoder” type of models. In fact, they are model-agnostic.
>
> - The conclusions derived, in addition to the transferability we discussed in the first paragraph, also inform the many use cases where LLMs are used to process relational data that haven’t been (or can’t be) encoded by a graph encoder. “LLM + graph encoder” is certainly an important architectural innovation and could be increasingly popular in graph-based RAG, where an explicit graph structure has to be ready in place as part of the prompt.
>
> **Question 2**
>
> Please see Figure 2, 3 in the new pdf uploaded. We group the tested graphs by the number of nodes they have, into three intervals: [5, 10), [10, 20), and [20, 30]. We then report accuracy (Figure 2) and bias scores (Figure 3) for graphs in each of the three groups. These two figures essentially provide more fine-grained results for Table 1 in our paper. We observe:
>
> - In terms of accuracy, we can see that larger graphs indeed have worse recall accuracy.
> - In terms of bias scores for microstructures, the strongest trend exists with "edge" patter, showing that LLMs forget significantly more edges for larger graphs. For other patters, we do not observe a strong consistent trend.
>
> **Question 3**
>
> Please refer to Section 2.1 for this. Briefly speaking, we fit a random graph model (ERGM) to the recalled graph output by LLMs. The model parameters ($\theta$) have the physical meaning of being the bias scores.
>
>
> **Question 4**
>
> Yes. Line 327 is explained by lines 328 - 332. Below are our further explanations.
>
>
> We observe that (1) LLMs tend to hallucinate triangles and “Alt-2-Paths” (i.e., the 5th pattern in Figure 2 Step 6) in both graph recall and link prediction tasks; (2) LLMs tend to hallucinate “Alt-Triangles” (i.e., the 4th pattern) in link prediction task, while forgetting this pattern in graph recall task.
>
>
> These indicate that LLMs do not always exhibit consistent biases between graph recall and link prediction: on some patterns, the bias across the two tasks is consistent; on other patterns, the bias across the two tasks can be opposite to each other.
>
>
> We also found this observation interesting or even counterintuitive to some extent, and that’s why we mentioned that we do not have a perfect explanation yet: because of triadic closure in link prediction, we had expected LLMs to favor less “Alt-Triangles” in link prediction. Our observation is the opposite, however. We had conjectured that the increase in “Alt-Triangles” might come from more spurious edges that get hallucinated but not from triadic closure over existing dense structures.
>
>
>
>
>
> **Question 5**
>
>
> Below please find our discussions of this interesting latest work (even though it came out after the submission deadline). We will also substantiate them further in our final version.
>
>
> [1]’s theory indicates that graph recall, as a fundamental graph reasoning task, should be solvable by a dedicated "small" transformer-based LLM. Meanwhile, our work shows through rigorous empirical tests that the general-purpose transformer-based LLMs achieve a performance much lower than the theoretical upper bound. Therefore, there exist huge opportunities for improvement. More concretely:
>
> - Theoretically, our proposed task of graph recall is indeed a graph retrieval task, which belongs to the simplest “D1” complexity class of problems for transformers. In other words, it is theoretically solvable by a “small” transformer (single-layer single-headed transformer with embedding dimension O(log N) where N is the length of the input node/edges sequence). This is consistent with our statement that graph recall is one of the simplest and most fundamental problems in graph reasoning.
> - Empirically, [1] shows in its Table 4 that there still exists a huge gap between the best empirical performance and theoretical upper bound. For example, even fine-tuned Palm has an accuracy of computing node degree < 0.7 most of the time. This suggests that there is still huge room and opportunities to improve LLM’s graph recall performance, and that it may not be necessary to switch to new architectures beyond transformers.
>
> Please let us know if we have addressed your concerns. Thank you!

---

> > ### Comment · Reviewer_QYAY · 2024-08-07
> >
> > Thanks for your response, which address my concerns. I have raised my score.

---

> > > ### Author Response · Authors · 2024-08-08
> > >
> > > Thank you. We are glad that your concerns have been addressed.

---

### Official Review · Reviewer_uAxF · 2024-07-12

**Soundness:** 2
**Presentation:** 2
**Contribution:** 2
**Rating:** 4
**Confidence:** 5

**Summary:**

The article discusses the issue of graph recall in large language models (LLMs) and conducts experiments on this topic. It tests the ability of LLMs to recall graphs, identifies factors influencing this ability, and examines the impact of this ability on subsequent tasks.

**Strengths:**

1.This article is the first to propose the scenario of graph recall. Existing methods do not yield good results for large models on graph tasks, making it difficult to identify the underlying issues.
2.It introduces an innovative method to verify the ability of graph recall. Using a heuristic approach, it evaluates the model's ability to recall graph structures.
3.The problem statement and method presentation are clear and well-organized.

**Weaknesses:**

1.The ability of graph recall should encompass many aspects, not just the recall of different micro structures.
2.The article mentions that the ability to recall is crucial for LLMs to complete graph tasks. However, the experimental section only examines the relationship between recall ability and link prediction.

**Questions:**

1.Are there other evaluation metrics to measure recall ability? Why can't direct edge prediction be considered a manifestation of recall?
2.For some nodes with rich text information, could this also be a manifestation of recall ability, potentially affecting prediction results?

**Limitations:**

1.The article provides limited explanation on how the graph recall ability of LLMs affects downstream tasks, focusing only on link prediction analysis.
2.Whether large models are sensitive to microstructures when processing graphs needs to be substantiated.

---

> ### Author Rebuttal · Authors · 2024-08-07
>
> We thank the reviewer for the review!
>
> **Weakness 1**
>
> We not only study the recall's microstructures, but also study:
> - the recall's accuracy (as the title suggests, and throughout the paper),
> - its correlation with other prediction tasks such as link prediction (Section 5) and node classification (new experiment, see Weakness 2),
> - its comparison with human behaviors (Section 3.3),
> - the many factors that can potentially affect it (Section 4).
>
> **Question 1**
> > Are there other evaluation metrics to measure recall ability?
>
> Please see "Weakness 2 & Limitation 1" below.
>
>
> > Why can't direct edge prediction be considered a manifestation of recall?
>
> Section 1, paragraphs 2-4 provide the answer. Here we summarize it as follows.
>
> We agree with the reviewer that edge prediction is related to graph recall, and we’ve conducted correlation studies on this (Section 5).
>
>
> However, edge prediction should not be considered as an inherent part of the graph recall task. Graph recall is a very different task from the downstream prediction tasks, including edge prediction, graph prediction, etc.  This is because the correct answer for graph recall always exists in the prompt and can be directly extracted, which is not true for any prediction task. Please see our definition of graph recall task in paragraph 3, which also follows references [8, 33, 24, 11, 41, 7, 21] that experimented with humans.
>
>
> We've followed the existing definition and focused on graph recall as a standalone task for studying LLMs because:
> graph recall is both the most straightforward and fundamental step for graph reasoning; as lines 38 - 40 explain, if an LLM can’t even remember what edges it has seen, it is likely to suffer greatly in those downstream graph reasoning tasks.
>
>
> Our experiment indeed shows LLMs suffer at graph recall, which is consistent with LLMs' poor performance at prediction tasks as well as other graph reasoning tasks [46, 34].
>
>
> Besides, the rich literature [8, 33, 24, 11, 41, 7, 21] in human studies also suggests that graph recall, under its current definition, is a scientifically meaningful and important topic to study.
>
> We have added an experiment on the correlation between graph recall and a node classification. Please see Figure 1 in our uploaded pdf.
>
>
> **Weakness 2 & Limitation 1**
>
>
> We have added a new experiment on the correlation between graph recall and node classification, please see Figure 1 in our uploaded pdf for more details. Our new experiment again shows that there exists positive correlation (r>0) between performance in graph recall and in node classification task, though the correlation is relatively less stronger. This is mainly due to the larger difference between the nature of the two tasks (graph recall and node classification).
>
> >The article mentions that the ability to recall is crucial for LLMs to complete graph tasks. However, the experimental section only examines ...
>
> This argument is based on the simple reasoning that graph recall is among the most straightforward and fundamental steps for graph reasoning:  If an LLM can’t even remember what edges it has seen, it is very likely to suffer in downstream graph reasoning tasks (note that this trend can also be empricially observed from plots in both correlation studies just discussed). Therefore, we need to understand LLM's graph recall first, before proceeding to downstream tasks.
>
>
> Besides, since LLM’s graph recall has never been studied, “relationship with downstream tasks” is only one of the many aspects of this topic that we need to examine (other aspects include accuracy, biased microstructures, comparison with humans, and factors that can affect it).
>
>
>
>
> **Question 2**
>
>
> We are not certain about the reviewer's definition of “rich text information”, so we have the following discussion.
>
>
> - If the reviewer considers semantic information of nodes as “rich text information”, then most of the nodes in our dataset are already described with concrete semantic meanings (Appendix D). In terms of results, which have been extensively discussed in Section 4.1, we've shown that different semantics in narration (i.e., what we call “narrative styles”) have interesting effects on recall accuracy.
>
>
> - If the reviewer has a different definition of “rich text information” (e.g., every node needs to have a feature vector or be associated with a long document), please let us know. We would be happy to discuss this further and/or provide more experiments.
>
>
> **Limitation 2**
>
>
> We are not certain about the exact meaning of LLMs being “sensitive to microstructures” as raised by the reviewer. Here are our answers based on two possible interpretations:
> - *Interpretation 1: “LLM’s graph recall performance is affected by microstructures of the input graph”.* We have not made such a claim in our paper, which is why it has not been “substantiated”. Here is further discussion: we conjecture this to be true because if we look at the microstructures of graph samples from different datasets (Figure 5 - 8 in Appendix), we can see that they have great variations. It does seem that either having too many cliques (e.g. the protein dataset) or too many spurious edges (or random graph dataset) can lead to low performance. We would be happy to provide more experiments if the reviewer is referring to this interpretation.
> - *Interpretation 2: “LLM’s recalled graphs have variations in microstructures”.* This is exactly one of the main themes which has been discussed throughout Section 3, with rigorous test procedures and numerical results that are statistically significant. If the reviewer could kindly let us know which particular statement needs to be substantiated, we would be happy to provide more discussion and/or experiment.
>
> Please let us know if we have addressed your concerns. Thank you!

---

> > ### Author Response · Authors · 2024-08-11
> >
> > Dear Reviewer,
> >
> > Please let us know if you have any further questions or concerns. Thank you very much for your time and consideration.
> >
> > Warm regards,
> >
> > Authors of Submission 4790

---

> ### Comment · Reviewer_uAxF · 2024-08-13
>
> Thanks for your rebuttal which has addressed some of my concerns, and thus I would like to raise the original score to reflect it.

---

> > ### Author Response · Authors · 2024-08-14
> >
> > Thank you for raising the score. We are glad to know that we have addressed your concerns.

---

### Official Review · Reviewer_iUa6 · 2024-07-13

**Soundness:** 3
**Presentation:** 4
**Contribution:** 3
**Rating:** 6
**Confidence:** 5

**Summary:**

This paper studies how well LLM models can recall graph structured information they have been provided with.  While the core of it is a fairly straightforward evaluation of LLM graph recall, it has an interesting experimental design inspired by psychology that adds substantially to the paper.  Some of the graphs used for evaluation come from real networks, which could bias the results (if, for example, an LLM's parametric knowledge had information about the nodes already).

**Strengths:**

+  interesting questions about recall ability of complex structure, with interesting results
+  interesting psychology inspired experimental design
+  nice presentation

**Weaknesses:**

-  I was left with some questions about disentangling parametric knowledge from recall (see questions below)
-  The (perhaps well intentioned) sex priming assignment experiment seems out of place with the rest of the work and could raise a yellow flag for some readers.  In its current form, the experiment and discussion adds little scientific value

**Questions:**

One of the biggest questions I have concerns how much parametric knowledge is being used for the "in-domain" narrative setting -- Are the node names random (when the "in-domain Graph Narratives" are used?) or e.g. are they actual proteins?  (if real protein names, are you assigning the correct name to each node?).  Related work on temporal graph reasoning (Test of Time: A Benchmark for Evaluating LLMs on Temporal Reasoning, https://arxiv.org/pdf/2406.09170) shows a significant effect from "masking" node ids.

Any results or discussion in this area would be interesting.

**Limitations:**

Yes

---

> ### Author Rebuttal · Authors · 2024-08-07
>
> We thank the reviewer for the review!
>
> **Weakness 1**
>
> Please see Question 1.
>
> **Weakness 2**
>
> We agree with the reviewer and will remove this mini-study or move it to the appendix.  As Section 4.3 explains, this mini-study was inspired by the previous human experiment “Sex and network recall accuracy” (our reference [7]). We included it for a comprehensive evaluation. However, the hypothesis was found to be not statistically significant.
>
> **Question 1**
>
> > Are the node names random (when the "in-domain Graph Narratives" are used?) or e.g. are they actual proteins?
>
> For the in-domain narrative setting, our datasets cover both cases: (1) where parametric knowledge is potentially useful, and (2) where parametric knowledge is less useful:
>
>
> - For protein networks, the protein names are not random. They are unique protein identifiers (known as the “UniProtKB/Swiss-Prot accession number” or NCBI index). Each node is assigned its real name. We also confirmed that LLMs know and precisely understand those protein identifiers. The same is true for DBLP coauthorship networks.
>
> - For all other networks, the node names are generated and randomly assigned. For example, in traffic (geographical) networks, the node names are “bank”, “townhall”, “high school”, etc.
>
>
> To further disentangle the effect of parametric knowledge and pure graph topology on graph recall, Section 4.1 introduces tests on cross-domain narratives (i.e. node names are randomly assigned from other domains or are plain indices like node 1, node 2, etc.). These tests essentially serve as an ablation study on parametric knowledge.
>
> > Any results or discussion in this area would be interesting.
>
> Regarding results and discussion in this area, we observe the following general trend in our numerical results:
>
> “in-domain narratives with real names ”  $\approx$ “in-domain narratives with permuted/generated names” > “cross-domain narratives”
>
>
> For example, see heatmaps in Figure 3 (a) – (c), where the diagonals are the in-domain cases; the off-diagonal cells show the cross-domain ablations.
>
>
> We observe that even when the LLM hasn’t seen the graph in training, they do better at recalling when the narrative style hints at the true domain from which the graph is sampled. This corresponds to the “no parametric knowledge” case. (or we might say the parametric knowledge factors in through a very implicit and subtle way, via LLM’s general understandings of the domain)
>
>
> That said, when the LLM happens to know about the graph to recall, parametric knowledge can certainly help with graph recall. We decided not to fully exclude this case in our test because many real-world graphs do come from domains familiar to LLMs.
>
>
> If the reviewer is particularly interested in the effect of parametric knowledge in protein networks, we would be happy to provide further experiments in which *real* protein names are *randomly* assigned to nodes.
>
> Please let us know if we have addressed your concerns. Thank you!

---

> > ### Author Response · Authors · 2024-08-11
> >
> > Dear Reviewer,
> >
> > Please let us know if you have any further questions or concerns. Thank you very much for your time and consideration.
> >
> > Warm regards,
> >
> > Authors of Submission 4790

---

> > ### Comment · Reviewer_iUa6 · 2024-08-13
> >
> > Thanks to to the authors for their reply, I have raised my score.
> >
> > > If the reviewer is particularly interested in the effect of parametric knowledge in protein networks, we would be happy to provide further experiments in which real protein names are randomly assigned to nodes.
> >
> > This would be an interesting experiment to add to the appendix if it isn't too costly to run.
> >
> > -  An additional thought while re-reading this paper -- recent theoretical results ("Understanding Transformer Reasoning Capabilities via Graph Algorithms", Sanford et al) categorize recall (in particular, edge recall iirc) as one of the simplest operations (it doesn't require a very "large" transformer network to perform it).  Any thoughts about how your results connect to this kind of theoretical framework would be insightful.

---

> > > ### Author Response · Authors · 2024-08-14
> > >
> > > Thank you for raising the score. We are glad that your concerns have been addressed.
> > >
> > > We will follow your advice to add the interesting experiment to appendix. Regarding the latest theoretical results, please see our response to Reviewer QYAY in Question 5. We will also include this in our final version.

---

### Author Rebuttal · Authors · 2024-08-07

Dear Reviewers,

We deeply appreciate your reviews. New experimental results have been provided in the attached pdf to this post. Our response to each reviewer has been posted separately.

We very much look forward to further interacting with you in the discussion period.

Warm regards,

Authors of Submission 4790

---

> ### Author Response · Authors · 2024-08-14
>
> Dear Reviewers and Meta Reviewer,
>
> Towards the end of the discussion period, we would like to sincerely thank you again for your reviews and discussion. We are glad to know that we have addressed concerns of the reviewers, who have all raised their scores. We will also update our final version to reflect all clarifications and suggested improvements.
>
> Warm regards,
>
> Authors of Submission 4790

---

### Decision · Program_Chairs · 2024-09-25

**Decision:**

Accept (poster)

**Comment:**

This paper studies the recall of micro-structures by various LLMs. While the approach is unorthodox for the NeurIPS community; I am positive that the paper will generate interesting discussion at the conference. I also note the positive change in the score by all 3 of the reviewers; it seems that most of the concerns are addressed in the rebuttal.